# The chloroplast genome sequence and phylogenetic analysis of *Apocynum venetum* L.

**Xiaonong Guo**[1,2,3]*, **Zhuanxia Wang**[1,2,3], **Deyu Cai**[1,2,3], **Lei Song**[1], **Jialin Bai**[1,2,3]

**1** Key Laboratory of Biotechnology and Bioengineering of State Ethnic Affairs Commission, Biomedical Research Center, Northwest Minzu University, Lanzhou, China, **2** College of Life Science and Engineering, Northwest Minzu University, Lanzhou, China, **3** China-Malaysia National Joint Laboratory, Biomedical Research Center, Northwest Minzu University, Lanzhou, China

* gxnwww@xbmu.edu.cn

**Data Availability Statement:** All relevant data are within the paper and its Supporting information files.

**Funding:** The National Natural Science Foundation of China (Grant No. 31760242), Gansu Provincial

## Abstract

*Apocynum venetum* L. (*Apocynaceae*) is valuable for its medicinal compounds and fiber content. Native *A. venetum* populations are threatened and require protection. Wild *A. venetum* resources are limited relative to market demand and a poor understanding of the composition of *A. venetum* at the molecular level. The chloroplast genome contains genetic markers for phylogenetic analysis, genetic diversity evaluation, and molecular identification. In this study, the entire genome of the *A. venetum* chloroplast was sequenced and analyzed. The *A. venetum* cp genome is 150,878 bp, with a pair of inverted repeat regions (IRA and IRB). Each inverted repeat region is 25,810 bp, which consist of large (LSC, 81,951 bp) and small (SSC, 17,307 bp) single copy areas. The genome-wide GC content was 38.35%, LSC made up 36.49%, SSC made up 32.41%, and IR made up 43.3%. The *A. venetum* chloroplast genome encodes 131 genes, including 86 protein-coding genes, eight ribosomal RNA genes, and 37 transfer RNA genes. This study identified the unique characteristics of the *A. venetum* chloroplast genome, which will help formulate effective conservation and management strategies as well as molecular identification approaches for this important medicinal plant.

## Introduction

*Apocynum venetum* L. (*Apocynaceae*) (Luobuma in Chinese) is a perennial herb distributed in Eurasia from Southeast Europe to Northern China. It occurs in floodplains and valleys along rivers such as the Tarim River [1, 2]. The roots, stems, leaves, and flowers of *A. venetum* have medicinal uses [3, 4] and these uses were documented in the "Compendium of Materia Medica." In 1977, *A. venetum* was listed in the Pharmacopoeia of the People's Republic of China as a primary treatment for hypertension and hyperlipidemia [5–8], and pharmacological studies have demonstrated that *A. venetum* possesses many pharmacological activities including cardiotonic [9], hepatoprotective [10, 11], antioxidant [12–14], antidepressant and anxiolytic effects [15–18]. *A. venetum* maybe useful for the prevention and treatment of cardiovascular and neurological diseases such as high blood pressure, high cholesterol, neurasthenia, depression, and anxiety [19–23].

Natural Science Foundation (Grant No. 20JR10RA120), Gansu Provincial Natural Science Foundation the Ministry of Education of China for an Innovative Research Team in University (IRT 17R88), and the Fundamental Research Funds for the Central Universities (Grant No. 31920190021) The funders had no role in study design, data collection and analysis, decision to publish, or preparation of the manuscript.

**Competing interests:** The authors have declared that no competing interests exist.

*A. venetum* has relatively high salt tolerance, cold tolerance, drought tolerance, high temperature tolerance, and wind resistance [24, 25]. It is an important plant for the wind proofing and sand-stabilization of desert grasslands in Central Asia. *A. venetum* therefore combines ecological benefits and economic benefits [24, 26]. Overharvesting of wild *A. venetum* and environmental degradation have reduced *Apocynum* populations and protection of *Apocynum* germplasm resources is needed. Studies of *A. venetum* have mainly focused on its medicinal effects and physiological characteristics such as photosynthesis and water absorption [27, 28]. However, there are few studies on the genetic diversity and genetic structure of wild *A. venetum* populations [29, 30].

Chloroplasts (cps) are the descendants of ancient bacteria endosymbionts. They are important organelles in plant cells that are responsible for photosynthesis and other aspects of metabolism [31]. Cp DNA is independent of the nuclear genome and exhibits semi-autonomous genetic characteristics. The characteristics of maternal and highly conserved genes in the cp genome are favorable for studying plant phylogeny [32, 33]. Molecular barcodes based on the cp genome have potential for species identification, especially among closely related taxa [34, 35]. The complete cp genome sequence may provide reliable barcodes for accurate plant identification at species and population levels [36, 37]. In higher plants, photosynthesis occurs in cp, which provides the necessary energy for plant growth and survival.

There are many counterfeit *A. venetum* products on the market, and they are difficult to detect based on appearance. There is a need for a molecular method to distinguish counterfeit products. DNA barcode sequence analysis is a molecular identification technology that uses standardized DNA sequence fragments to provide a fast, accurate, and automated species identification method [38–41]. The non-coding region of the cp has been successfully used in research on the DNA barcode. *A. venetum* cp genome information can provide candidate DNA barcodes for the identification of *A. venetum* and counterfeit products.

In this study, we assembled and analyzed the *A. venetum* cp genome sequence based on Illumina paired-end (PE) sequencing data. Through bioinformatics analysis, the sequence was compared with other known cp genome sequences. The information helped us determine the phylogeny of this species.

## Materials and methods

### Sampling, DNA extraction, sequencing, and assembly

*A. venetum* seeds were collected from wild plants in Shaya County in the Xinjiang Uygur Autonomous Region, China (40˚92´N, 82˚21´E; 957 m). After removal of the bracts, seeds were surface sterilized for 1 min in 75% ethanol (v/v), rinsed three times with distilled water, and then germinated at 25˚C in the dark on filter paper dampened with distilled water. When the plumule emerged, uniform seedlings were transplanted into plugged holes in plastic containers (5 cm × 5 cm × 5 cm, 1 seedling/container) filled with vermiculite and watered with modified Hoagland nutrient solution containing 2 mM $KNO_3$, 0.5 mM $NH_4H_2PO_4$, 0.25 mM $MgSO_4 \cdot 7H_2O$, 0.1 mM $Ca(NO_3)_2 \cdot 4H_2O$, 50 μM Fe-citrate, 92 μM $H_3BO_3$, 18 μM $MnCl_2 \cdot 4H_2O$, 1.6 μM $ZnSO_4 \cdot 7H_2O$, 0.6 μM $CuSO_4 \cdot 5H_2O$ and 0.7 μM $(NH_4)_6Mo_7O_{24} \cdot 4H_2O$. Solutions were renewed every 3 d. Seedlings were grown in a greenhouse at a temperature of 28˚C/23˚C (day/night) and photoperiod of 16:8 h (light:dark). The flux density was approximately 800 μmol $m^{-2}$ $s^{-1}$) and the relative humidity was 65%. Fresh leaves were collected on October 18, 2019, frozen in liquid nitrogen and then stored at −80˚C until analysis [42].

Genomic DNA was isolated by the modified CTAB method. Agarose gel electrophoresis and a one drop spectrophotometer (OD-1000, Shanghai, China) were used to detect DNA integrity and quality. One library (250 bp) was constructed using pure DNA according to the

manufacturer's instructions (NEBNext® UltraTM DNA Library Prep Kit for Illumina®). The library was constructed with an Illumina NovaSeq platform (Benagen Tech Solution Co. Ltd., Wuhan, China) and 150-bp paired-end reads were generated. The Illumina PCR adapter reads, low-quality reads and reads containing more than 5% unknown nucleotides "Ns" were filtered from the paired-end raw reads in the quality control step. All good-quality paired clean reads were obtained using SOAPnuke software (version: 1.3.0). The assembled reads were joined into a bidirectional iterative derivation using NOVOPlasty (version:3.13.1, parameter: k-mer = 127) to obtain the whole-genome sequence. The cp-like reads were used to assemble sequences using NOVOPlasty. NOVOPlasty assembled the partial reads and stretched as far as possible until a circular genome was formed. All circled sequences were searched by BLASTN (version: BLAST 2.2.30+, E-value $\leq 1^{e-5}$) against the reference database. Sequences with alignment greater than 1,000 bp and coverage greater than 90% were retained. Based on the depth of sequencing, PE reads alignment, and alignment with closely species to *A. venetum*, the candidate sequences were connected in order to determine whether they formed a loop. When a gap (including N sequence) appeared, Gapcloser (Version: 1.12) was used to fill in the hole to obtain the final splicing result [43]. After filtering the repeated sequences and the sequences with lengths less than 300 bp, 48 sequences with start codons of ATG, TTG, CTG, ATT, ATC, GTG, and ATA and end codons of TGA, TAG, and TAA, were retained to conduct subsequent analysis.

## Annotation and analysis of the cpDNA sequences

The cp genome sequence was annotated using the DOGMA program (http://dogma.ccbb. utexas.edu/) [44], and the tRNAscan-SE program was used to predict tRNAs in the genome [45]. The circular maps were drawn by the OGDRAWv1.2 program [46] (http://ogdraw. mpimp-golm.mpg.de/). In order to eliminate the influence of amino acid composition on codon usage, the characteristics of the variations in synonymous codon usage, the relative synonymous codon usage values (RSCU), base composition and codon content were analyzed using MEGA 7.0. Simple sequence repeats (SSRs) in the cp genome were identified using SSRHunter software (http://www.biosoft.net) [47, 48]. The parameters were set to five repeat units for mononucleotide SSRs, five repeat units for dinucleotide SSRs, three repeat units for trinucleotide SSRs, and three repeat units each for tetranucleotides, and pentanucleotide SSRs.

## Genome comparison

The pairwise alignments of cp genomes was conducted by MUMmer [42]. The mVISTA software was used to compere the *A. venetum* cp genome with three other cp genomes. *Nicotiana attenuata*, *Gossypium hirsutum*, and *Arabidopsis thaliana* (NC_035952.1, DQ345959, and NC_000932.1, respectively) using the annotation of *Sophora japonica* L. as reference [44, 45]. We determined the repeat structure, including forward and reverse repeats, using the REPuter software [46–49].

## Phylogenetic analysis

We downloaded 21 cp genome sequences from the NCBI organelle genome and nucleotide resource database, and used all genomes for phylogenetic analysis. Clustalw2 software (Conway Institute of Biomolecular and Biomedicine, Dublin, Ireland) was used to sequence the genome [50–53]. We used MEGA7.0 to analyze and draw a phylogenetic tree with ML (maximum likelihood). Bootstrap analysis was performed using 1,000 repetitions and TBR branch exchanges [54–56]. We used 1,000 replicates and TBR branch exchanges to complete the bootstrap analysis.

## Results

### Features of *A. venetum* cpDNA

The complete cp genome of *A. venetum* is 150,878 bp in length (GenBank accession number: MT568765) (Fig 1), and includes a pair of inverted repeats (IR) 25,810 bp long, separated by a large single region (LSC) and a small copy region (SSC) of 81,951 bp and 17,307 bp, respectively (Table 1). It is similar to the cp genome of other *Apocynaceae* species [57]

In the *A. venetum* cp genome, 131 functional genes were predicted, including eight rRNA genes, 37 tRNA genes, and 86 protein-coding genes (Table 2) Cp genomes in the IR regions include 33 duplicated genes, with approximately 15 tRNA genes (tRNAs), eight rRNA genes (rRNAs), and nine protein-coding genes (PCGs) (Fig 1). The LSC region includes 58 protein-

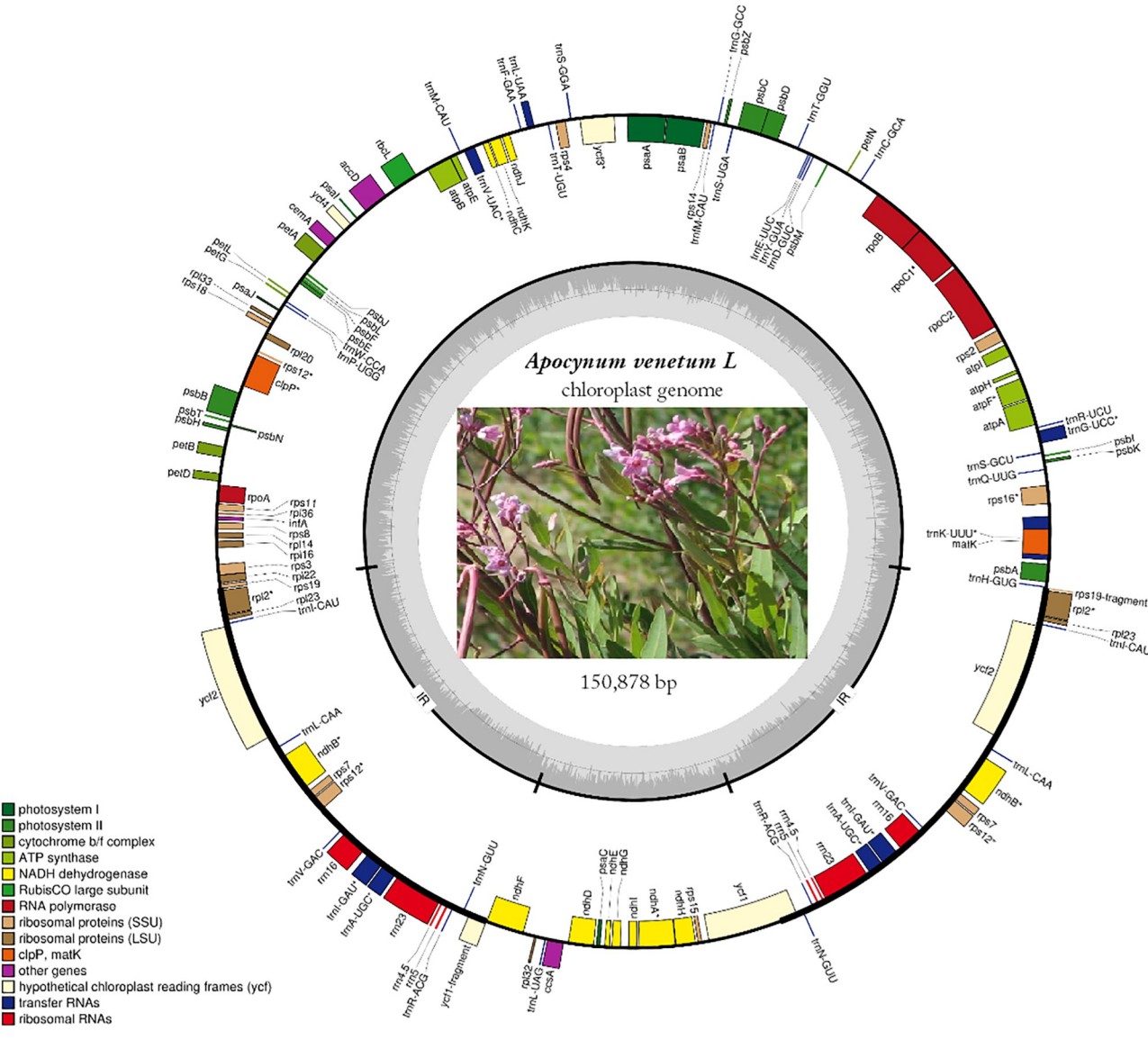

**Fig 1. Map of *A. venetum* cpgenome.** Thick lines indicate the extent of the inverted repeat regions (Ira and Irb), which separate the genome into small (SSC) and large (LSC) single copy regions. Genes drawn inside the circle are transcribed clockwise, and those outside are transcribed counterclockwise. Different colors represent different functional groups of genes.

**Table 1. Base composition in the *A. venetum* chloroplast genome.**

| Region | Length | A (%) | T (%) | C (%) | G (%) | AT (%) | GC (%) |
|---|---|---|---|---|---|---|---|
| Total_genome | 150878 | 30.43 | 31.21 | 19.52 | 18.83 | 61.64 | 38.35 |
| LSC | 81951 | 31.02 | 32.49 | 18.69 | 17.8 | 63.51 | 36.49 |
| IRA | 25810 | 28.59 | 28.11 | 20.84 | 22.46 | 56.7 | 43.3 |
| SSC | 17307 | 33.85 | 33.65 | 17.09 | 15.32 | 67.49 | 32.41 |
| IRB | 25810 | 28.11 | 28.59 | 22.46 | 20.84 | 56.7 | 43.3 |

coding and 22 tRNA genes, while the SSC region includes one tRNA gene and 11 protein-coding genes.

The tRNA and protein-encoding gene sequences of the *A. venetum* cp were analyzed, and the codon usage frequency of the cp genome of *A. venetum* was inferred and summarized. A total of 17,318 codons represent the coding ability of 86 protein-coding genes and tRNA genes of *A. venetum* (Table 4), of which 1,814 codons code for leucine (10.47%), and 319 codons code for tryptophan (1.84%), which are the most common and least common amino acids in the cp genome of *A. venetum*, respectively. Codons ending in A and U are very common. Except for trtl-caa, all preferred synonymous codons (RSCU > 1) end in A or U. There are 14 intron-containing genes, including nine protein-coding genes and five tRNA genes (Table 3). Twelve genes (seven protein-coding and five tRNA genes) contain an intron, and two genes (ycf3 and clpP) contain two introns of the intragene region (Table 3). The size of the intron-

**Table 2. Genes present in the *A. venetum* chloroplast genome.**

| Category for genes | Group of genes | Name of genes |
|---|---|---|
| Transcription and translation-related genes | transfer RNAs | trnM-CAU, trnR-ACG, trnY-GUA, trnG-UCC, trnL-UAG, trnI-GAU, trnW-CCA, trnR-UCU, trnQ-UUG, trnL-UAA, trnS-GGA, trnH-GUG, trnT-GGU, trnT-UGU, trnP-UGG, trnK-UUU, trnN-GUU, trnG-GCC, trnI-CAU, trnD-GUC, trnF-GAA, trnS-GCU, trnS-UGA, trnfM-CAU, trnE-UUC, trnV-GAC, trnA-UGC, trnV-UAC, trnL-CAA, trnC-GCA |
| | RNA polymerase | rpoB, rpoA, rpoC1, rpoC2 |
| | ribosomal proteins(SSU) | rps8, rps4, rps16, rps14, rps7, rps12, rps2, rps11, rps19-fragment, rps19, rps18, rps3, rps15 |
| | ribosomal proteins(LSU) | rpl2, rpl23, rpl32, rpl33, rpl36, rpl14, rpl16, rpl22, rpl20 |
| | Translational initiation factor | infA |
| | ribosomal RNAs | rrn4.5, rrn5, rrn23, rrn16 |
| Photosynthesis-related genes | NADH dehydrogenase | ndhA, ndhH, ndhF, ndhJ, ndhE, ndhI, ndhG, ndhK, ndhC, ndhD, ndhB |
| | photosystem I | psaI, psaJ, psaC, psaB, psaA |
| | photosystem II | psbA, psbL, psbF, psbB, psbK, psbJ, psbM, psbT, psbE, psbD, psbC, psbH, psbI, psbN, psbZ |
| | cytochrome b/f complex | petL, petN, petB, petG, petA, petD |
| | RubisCO | rbcL |
| | ATP synthase | atpA, atpE, atpH, atpI, atpB, atpF |
| | hypothetical chloroplast reading frames(ycf) | ycf2, ycf4, ycf1, ycf3, ycf1-fragment |
| Other genes | Maturase | matK |
| | Protease | clpP |
| | Envelope membrane protein | cemA |
| | Subunit of Acetyl-CoA carboxylase | accD |
| | C-type cytochrome synthesis gene | ccsA |

**Table 3. Length of exons and introns in genes with introns in the *A. venetum* chloroplast genome.**

| Gene | Location | Exon I (bp) | Intron I (bp) | Exon II (bp) | Intron II (bp) | Exon III (bp) |
|------|----------|-------------|---------------|--------------|----------------|---------------|
| trnK-UUU | LSC | 35 | 2474 | 37 | | |
| rps16 | LSC | 226 | 837 | 41 | | |
| trnG-UCC | LSC | 23 | 672 | 48 | | |
| atpF | LSC | 411 | 706 | 144 | | |
| rpoC1 | LSC | 1599 | 748 | 451 | | |
| ycf3 | SSC | 155 | 794 | 226 | 717 | 126 |
| trnV-UAC | LSC | 37 | 588 | 36 | | |
| rps12 | LSC | 114 | 536 | 234 | | |
| clpP | LSC | 228 | 642 | 291 | 763 | 69 |
| rpl2 | IR | 434 | 649 | 391 | | |
| ndhB | IR | 756 | 685 | 777 | | |
| trnI-GAU | IR | 37 | 952 | 35 | | |
| trnA-UGC | IR | 38 | 817 | 35 | | |
| ndhA | SSC | 545 | 1039 | 553 | | |

containing matK gene in the trnK-UUU gene was 2,474 bp. The Rps12 gene is a trans-splicing gene with the 5' end in the LSC region and the 3' end in the IR region.

## Comparative analysis of genomic structure

Comparative genome analysis permits the examination of how DNA sequences diverge among related species. The whole cp genome sequence of *A. venetum* was compared to the sequences of *N. attenuata*, *G. hirsutum*, and *A. thaliana*. The identities of the entire sequence of the four cp genomes were drawn using the annotation mVISTA *N. attenuata* as a reference (Fig 2). The variation of the LSC and SSC regions were significantly greater than that of the IR regions. Moreover, the coding regions were more conserved than the non-coding regions. The most divergent coding regions of the four cp genomes were rnH-psbA, psbM-petN, trnC-GCA-petN, trnE-UUC-rpoB, trnY-GUA-trnE-UUC, trnV-UAC-ndhC, rbcL-accD, accD- psaI, LSC rpl32-trnL-UAG, and ndhI-ndhG ycf1-rps15 SSC, and the distribution of plastid rRNAs (rrn4.5, rrn5, rrn16, and rrn23) was the most conserved.

## Repeat sequence analysis

We studied the type, existence, and distribution of SSR in the cp genome of *A. venetum*. A total of 273 SSRs were found in *A. venetum*, most of which were distributed in LSC and SSC, and some in IR. These included 105 single nucleotide SSRs (38.46%), 142 dinucleotide SSRs (50.01%), 10 trinucleotides, 14 tetranucleotides, and two pentanucleotide repeats. The mono-nucleotide A and T repeat units accounted for the largest portion.

## Phylogenetic analysis

The cpDNA gene content is highly conserved in most land plants. We downloaded 21 complete cp genome sequences from the NCBI Organelle Genome Resources database to reveal the phylogenetic location of *A. venetum* (Fig 3). In this study, we constructed a phylogenetic tree to infer the phylogenetic positions of *A. venetum* cp genomes. The evolutionary tree was separated into four clusters. The phylogenetic tree showed that *Vitis vinfera* were clustered on a single terminal branch. Phylogeny analysis showed that *Glycine max*, *Ricinus communis*, *Populus trichocarpa*, *Prunus persica*, *Medicago truncatuta*, *Capsella rubella*, *A. thaliana*, and

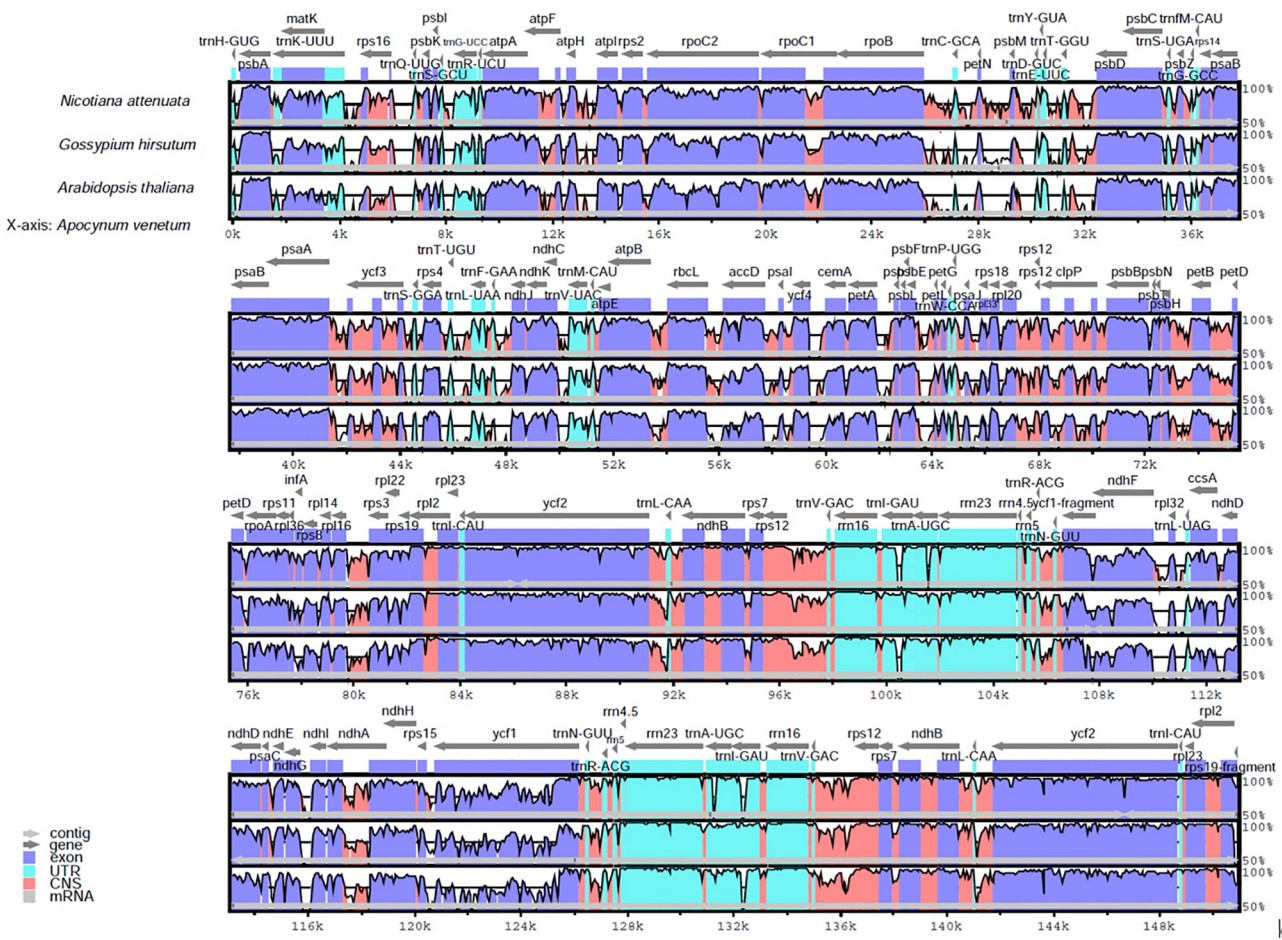

**Fig 2. Comparison of the cpgenome sequences of four plants.** Comparison of the cp genome sequences of *N attenuata*, *G. hirsutum*, *A. thaliana*, and *A. venetum* generated with mVISTA. Gray arrows indicate the position and direction of each gene. Red and blue areas indicate the intergenic and genic regions, respectively. The vertical scale indicates the percentage of identity, ranging from 50% to 100%.

*Eutrema salsgineum* formed an independent branch. We found that *A. venetum* L. was grouped into a terminal branch with *Lonicra japonica* and *N. atienuate*, *Capsicum annuum*, *Solanum tuberosum*, *Solanum lycopersicum* and *Salicornia europaea*. Meanwhile, *Nelumbo nucifera*, *Poenix dactylifera*, *Zea mays*, *Triticum aestivum*, and *Hordeum vulgare* were clustered on a branch.

## Discussion

In this study, we assembled, annotated and analyzed the complete cp sequence of *A. venetum*. We then analyzed its features, GC content, gene structure, and repeat sequences. The complete cp genome of *A. venetum* has a total length of 150,878 bp, with a pair of IRs of 25,810 bp that separate an LSC region of 81,951 bp and an SSC region of 17,307 bp. The DNA GC content of LSC, SSC, IR, and the whole genome were 36.49%, 32.41%, 43.3%, and 38.36%, respectively, which were similar to those of other species in *Nerium*. DNA GC content is an important index to evaluate the genetic relationship of *Nerium oleracea*, and the cpDNA GC content of *Nerium indicum* is similar to that in other species of *Apocynaceae* [58–63]. The content of DNA GC in the IR region is higher than that in other regions (LSC, SSC); this phenomenon is

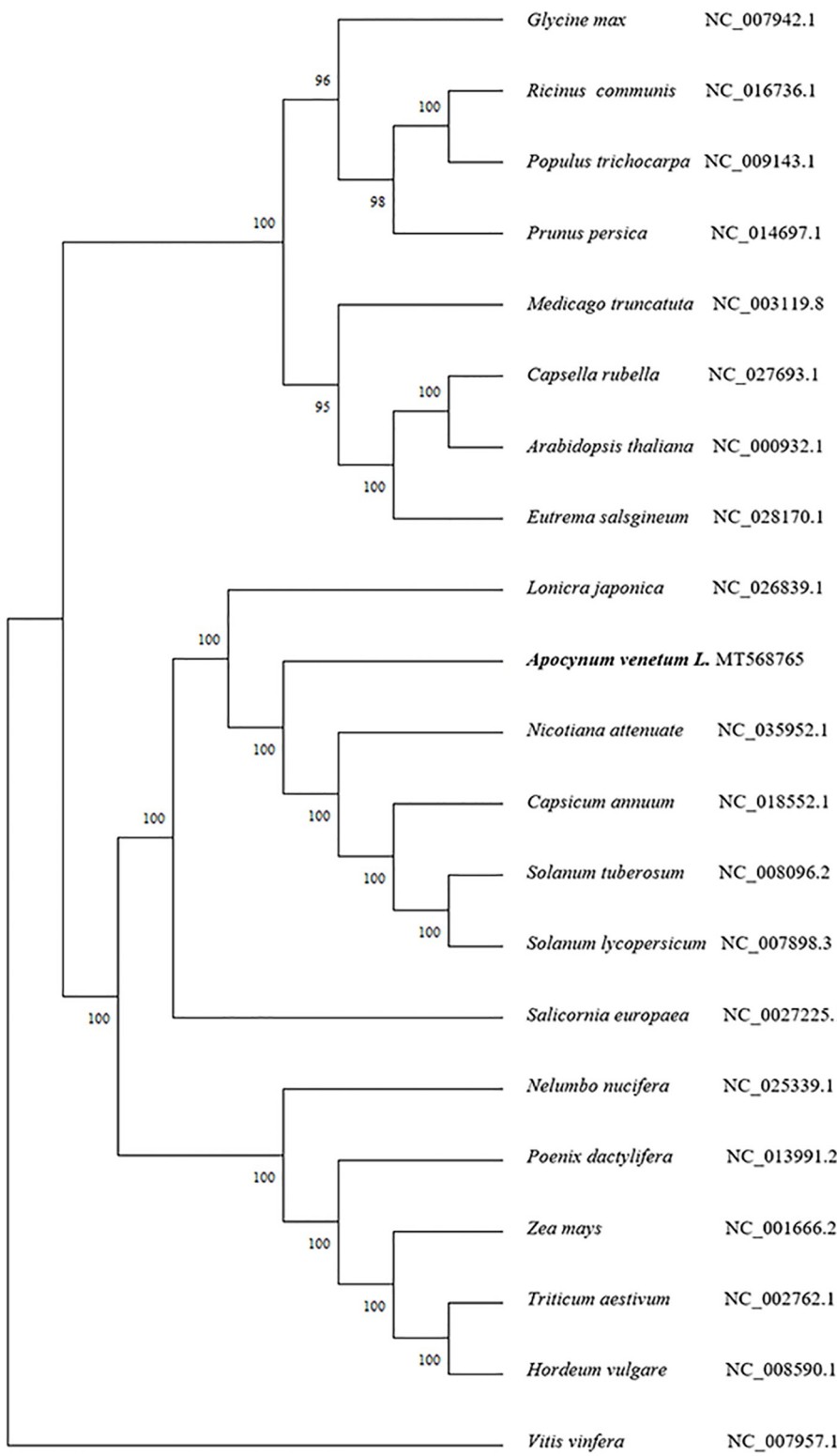

**Fig 3. Phylogenetic tree analysis of whole chloroplast genome.** Maximum likelihood (ML) phylogenetic tree reconstruction including 21 species based on all chloroplast genomes. The bootstrap value, based on 1,000 replicates, is shown on each node. *V. vinfera* was used as the outgroup. The GenBank accession numbers are listed following the species name.

common in other plants [64–66]. The relatively high DNA GC content in the IR region was mainly attributed to the rRNA gene and the tRNA gene [67, 68].

Cp sequences have been used to compare the genetics of plant species, gene flow between species, and the size of ancestral populations of sister species [69]. Therefore, it is necessary to understand cp differences among species. We observed the order of approximately the same genes and the coding regions in the organization of the cp genome (Fig 2). The cp genome is considered to be highly conservative compared to the non-coding region, and the two infrared regions are less divergent than the LSC and SSC regions. The four cp genomes with the most different coding regions (rnH-psbA, psbM-petN, trnC-GCA-petN, trnE-UUC-rpoB, trnY--GUA-trnE-UUC, trnV-UAC-ndhC, rbcL-accD, accD-psaI, LSC rpl32-trnL-UAG, and ndhI-ndhG ycf1-rps15 SSC) and the four ribosomal RNA genes (rrn4.5, rrn5, rrn16, and rrn23) were the most conserved. Similar results have been observed in other plant cp genomes.

**Table 4. Codon-anticodon recognition patterns and codon usage of the *A. venetum* chloroplast genome.**

| Amino Acid | Codon | Number | RSCU* | tRNA | Amino Acid | Codon | Number | RSCU* | tRNA |
|---|---|---|---|---|---|---|---|---|---|
| Stop | UAA | 23 | 1.53 | | Met | AUG | 401 | 1 | trnM-CAU |
| Stop | UAG | 9 | 0.6 | | Asn | AAU | 627 | 1.56 | |
| Stop | UGA | 13 | 0.87 | | Asn | AAC | 178 | 0.44 | |
| Ala | GCU | 459 | 1.78 | | Pro | CCU | 270 | 1.49 | |
| Ala | GCC | 174 | 0.67 | | Pro | CCC | 153 | 0.84 | |
| Ala | GCA | 281 | 1.09 | | Pro | CCA | 191 | 1.05 | trnP-UGG |
| Ala | GCG | 118 | 0.46 | | Pro | CCG | 111 | 0.61 | |
| Cys | UGU | 128 | 1.44 | | Gln | CAA | 495 | 1.54 | trnQ-UUG |
| Cys | UGC | 50 | 0.56 | trnC-GCA | Gln | CAG | 148 | 0.46 | |
| Asp | GAU | 578 | 1.59 | | Arg | CGU | 227 | 1.32 | trnR-ACG |
| Asp | GAC | 147 | 0.41 | trnD-GUC | Arg | CGC | 78 | 0.45 | |
| Glu | GAA | 682 | 1.51 | trnE-UUC | Arg | CGA | 240 | 1.39 | |
| Glu | GAG | 220 | 0.49 | | Arg | CGG | 90 | 0.52 | |
| Phe | UUU | 611 | 1.28 | | Arg | AGA | 285 | 1.65 | trnR-UCU |
| Phe | UUC | 343 | 0.72 | trnF-GAA | Arg | AGG | 115 | 0.67 | |
| Gly | GGU | 412 | 1.28 | | Ser | UCU | 366 | 1.67 | |
| Gly | GGC | 144 | 0.45 | trnG-GCC | Ser | UCC | 223 | 1.02 | trnS-GGA |
| Gly | GGA | 474 | 1.48 | | Ser | UCA | 246 | 1.12 | trnS-UGA |
| Gly | GGG | 253 | 0.79 | | Ser | UCG | 131 | 0.6 | |
| His | CAU | 315 | 1.45 | | Ser | AGU | 269 | 1.23 | |
| His | CAC | 120 | 0.55 | trnH-GUG | Ser | AGC | 81 | 0.37 | trnS-GCU |
| Ile | AUU | 722 | 1.5 | | Thr | ACU | 353 | 1.63 | |
| Ile | AUC | 294 | 0.61 | trnI-CAU | Thr | ACC | 176 | 0.81 | trnT-GGU |
| Ile | AUA | 431 | 0.89 | | Thr | ACA | 238 | 1.1 | trnT-UGU |
| Lys | AAA | 594 | 1.49 | | Thr | ACG | 98 | 0.45 | |
| Lys | AAG | 204 | 0.51 | | Val | GUU | 361 | 1.49 | |
| Leu | UUA | 597 | 1.97 | | Val | GUC | 114 | 0.47 | trnV-GAC |
| Leu | UUG | 365 | 1.21 | trnL-CAA | Val | GUA | 360 | 1.48 | |
| Leu | CUU | 388 | 1.28 | | Val | GUG | 135 | 0.56 | |
| Leu | CUC | 107 | 0.35 | | Trp | UGG | 319 | 1 | trnW-CCA |
| Leu | CUA | 225 | 0.74 | | Tyr | UAU | 512 | 1.64 | |
| Leu | CUG | 132 | 0.44 | | Tyr | UAC | 114 | 0.36 | trnY-GUA |

RSCU *: relative synonymous codon usage.

Cp genomes are highly conserved and contain a large amount of genetic information. The noncoding regions are less conserved than the coding regions [70, 71]. The genes trnK-UUU, rps16, trnG-UCC, atpF, rpoC1, trnV-UAC, rps12, rpl2, ndhB, trnI-GAU, trnA-UGC, and ndhA have one intron each, while clpP and ycf3 contain two introns. A trans-splicing event was also observed in the rps12 gene (Table 4). Previous studies have reported that ycf3 is necessary for the stable accumulation of photosystem I complexes [42, 72]. Therefore, we believe that the intron gain in ycf3 of *A. venetum* provides insight into the evolution of photosynthesis. As cp-specific SSRs are inherited from one parent and are mainly formed by the chain mismatch caused by the sliding of polymerase during DNA replication, they are often used in population genetics, species identification, and evolutionary process research on wild plants. In addition, the cp genome sequence is highly conserved, and SSR primers of cp genome can be transferred across species and genera. There were 273 SSRs detected in in the CP genome of *A. venetum*. Among these SSRs, mono-, di-, tri-, tetra-, and pentanucleotide were detected. The average density of SSRs was 1.809 SSR/kb in *A. venetum* (A/T as the main component). These cpSSR markers could be used for future studies of the genetic structure, diversity, and differentiation of *A. venetum* and its related species.

The phylogenetic positions of 21 cp genomes were successfully analyzed with the support of full bootstrap at almost all nodes. A phylogenetic tree was constructed for the data by ML, and *V. vinfera* was used as an outgroup. In this method, an initial tree is first built using a fast but suboptimal method such as the neighbor-joining method, and its branch lengths are adjusted to maximize the likelihood of the data set for that tree topology under the desired model of evolution. The results show that *A. venetum* has the closest relationship with *L. japonica*, *N. attenuata*, *C. annuum*, *S. tuberosum*, and *S. lycopersicum*.

## Conclusion

We analyzed and illustrated the complete cp genome of *A. venetum*. The cp genome is conservative and similar to other species of *Apocynum*. These results provide a reference for the complete assembly of the cp genome of *Apocynaceae*, which may aid future breeding and research efforts. It may also assist in the development of unique *Apocynaceae* DNA barcodes of *Apocynaceae* and in determining the evolutionary history of *Apocynaceae*.

## Supporting information

**S1 File.**
(ZIP)

**S2 File.**
(RAR)

**S3 File.**
(ZIP)

## Acknowledgments

We thank LetPub (www.letpub.com) for its linguistic assistance during the preparation of manuscript.

## Author Contributions

**Conceptualization:** Xiaonong Guo, Zhuanxia Wang.

**Data curation:** Xiaonong Guo, Deyu Cai, Lei Song, Jialin Bai.

**Formal analysis:** Xiaonong Guo.

**Funding acquisition:** Xiaonong Guo.

**Methodology:** Xiaonong Guo.

**Project administration:** Xiaonong Guo.

**Resources:** Xiaonong Guo.

**Software:** Xiaonong Guo, Zhuanxia Wang, Deyu Cai.

**Writing – original draft:** Xiaonong Guo, Zhuanxia Wang, Deyu Cai, Lei Song, Jialin Bai.

**Writing – review & editing:** Xiaonong Guo.

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
