## [Decision Letter · Decision Letter 0]

25 Aug 2021

PONE-D-21-20974

The chloroplast genome sequence and phylogenetic analysis of Apocynum venetum L.

PLOS ONE

Dear Dr. Wang,

Thank you for submitting your manuscript to PLOS ONE. After careful consideration, we feel that it has merit but does not fully meet PLOS ONE’s publication criteria as it currently stands. Therefore, we invite you to submit a revised version of the manuscript that addresses the points raised during the review process.

We look forward to receiving your revised manuscript.

Kind regards,

Maoteng Li

Academic Editor

PLOS ONE

“The National Natural Science Foundation of China (Grant No. 31760242), Gansu Provincial Natural Science Foundation (Grant No. 20JR10RA120), Gansu Provincial Natural Science Foundation the Ministry of Education of China for an Innovative Research Team in University (IRT 17R88), and the Fundamental Research Funds for the Central Universities (Grant No. 31920190021)”.

We note that you have provided funding information that is not currently declared in your Funding Statement. However, funding information should not appear in the Funding section or other areas of your manuscript. We will only publish funding information present in the Funding Statement section of the online submission form.

“The funders had no role in study design, data collection and analysis, decision to publish, or preparation of the manuscript”.

5. PLOS requires an ORCID iD for the corresponding author in Editorial Manager on papers submitted after December 6th, 2016. Please ensure that you have an ORCID iD and that it is validated in Editorial Manager. To do this, go to ‘Update my Information’ (in the upper left-hand corner of the main menu), and click on the Fetch/Validate link next to the ORCID field. This will take you to the ORCID site and allow you to create a new iD or authenticate a pre-existing iD in Editorial Manager. Please see the following video for instructions on linking an ORCID iD to your Editorial.

6. Please amend the manuscript submission data (via Edit Submission) to include author Xiaonong Guo.

7. Thank you for submitting the above manuscript to PLOS ONE. During our internal evaluation of the manuscript, we found significant text overlap between your submission and the following previously published works, some of which you are an author.

https://www.researchgate.net/publication/322729499_Complete_Chloroplast_Genome_Sequence_and_Phylogenetic_Analysis_of_Paeonia_ostii

https://www.mdpi.com/1422-0067/19/8/2443/htm

Please revise the manuscript to rephrase the duplicated text, cite your sources, and provide details as to how the current manuscript advances on previous work. Please note that further consideration is dependent on the submission of a manuscript that addresses these concerns about the overlap in text with published work.

Reviewers' comments:

Reviewer's Responses to Questions

**Comments to the Author**

1. Is the manuscript technically sound, and do the data support the conclusions?

Reviewer #1: Partly

Reviewer #2: Yes

2. Has the statistical analysis been performed appropriately and rigorously? 

Reviewer #1: N/A

Reviewer #2: Yes

3. Have the authors made all data underlying the findings in their manuscript fully available?

Reviewer #1: Yes

Reviewer #2: Yes

4. Is the manuscript presented in an intelligible fashion and written in standard English?

Reviewer #1: Yes

Reviewer #2: No

5. Review Comments to the Author

Reviewer #1: In this manuscript, the authors reported a chloroplast genome sequence Apocynum venetum and performed phylogenetic analysis of this species with other land plants. However, I do have some concerns before its final acceptance.

1. While the study appears to be sound, the language is unclear, making it difficult to follow. I advise the authors work with a writing coach or copyeditor to improve the flow and readability of the text.

2. The authors should carefully read and follow the PLOS ONE guidelines. There is no line number in the manuscript. Figures should be cited “Fig 1”, “Fig 2”, etc. References with more than six authors should list the first six author names, followed by “et al.”

3. The authors should give more details for chloroplast genome assembly. How they defined the chloroplast-like reads? Please check the version of NOVOPlasty.

4. In Results, there are no results description in “Comparative Analysis of Genomic Structure” and “Phylogenetic Analysis”.

5. In Discussion, authors should avoid the first paragraph, they are part of the Introduction. And, authors need to emphasize the novel insights obtained from their study.

6. The authors said “used 13 chloroplast genome sequences for phylogenetic analysis” in Page 5, but there are 21 complete chloroplast genome sequences in Page 10 and Figure 3. Please check and correct it.

7. Which software was used to draw the phylogenetic tree? The authors said “used MEGA7.0 to analyze and draw a phylogenetic tree with ML (maximum likelihood)” in method. However, in Discussion was “the Bayesian Inference (BI) method based on RAxML was used to construct a phylogenetic tree”. Please carefully check.

8. The reviewer suggested the authors to remove the Table 2.

9. In page 12, the authors mentioned the reference 74 in “Previous studies have reported that ycf3 is necessary for the stable accumulation of photosystem I complexes [42, 74]”, but missed it in References.

Reviewer #2: This manuscript “The chloroplast genome sequence of Apocynum venetum L.” reported the chloroplast genome sequence of Apocynum venetum, and provided some basic information about this genome, such as GC contents, LSC, SSC and coding gene information, which will be helpful for the studies in plant evolution. However, the written is very poor, and it is very descriptive that missing some very important information. For example, the author claim that the A. venetum chloroplast genome encodes 131 genes, including 86 protein-coding genes, 8 ribosomal RNA genes, and 37 transfer RNA genes. Is there any novel genes? How about this gene expression pattern, especially when plants meet the stress? The author could also compare the protein amount of chloroplast genes in some relative plant species and try to explain why Apocynum venetum could grow in such dry condition.

6. PLOS authors have the option to publish the peer review history of their article (what does this mean?). If published, this will include your full peer review and any attached files.

Reviewer #1: No

Reviewer #2: No

---

## [Author Response · Author response to Decision Letter 0]

9 Oct 2021

Dear Editors and Reviewers:

Thank you for your advice and for the reviewers’ comments concerning our manuscript entitled “The chloroplast genome sequence and phylogenetic analysis of Apocynum venetum L”. Those comments are all valuable and very helpful for revising and improving our paper, in addition to their important guiding significance for our research. We have studied the comments carefully and have made corrections, which we hope will meet with approval. The revised portions are marked in red in the paper. The main corrections in the paper and the responses to the editor’s and reviewers’ comments are as follows:

Responses to the editor’s comments:

1. Please ensure that your manuscript meets PLOS ONE's style requirements, including those for file naming. The PLOS ONE style templates can be found at https://journals.plos.org/plosone/s/file?id=wjVg/PLOSOne_formatting_sample_main_body.pdf and https://journals.plos.org/plosone/s/file?id=ba62/PLOSOne_formatting_sample_title_authors_affiliations.pdf.

Response: Thank you very much for your comments, which helped to improve our manuscript. We have revised the manuscript accordingly.

Response: Thank you very much for your comments. We have revised the mistakes.

“The National Natural Science Foundation of China (Grant No. 31760242), Gansu Provincial Natural Science Foundation (Grant No. 20JR10RA120), Gansu Provincial Natural Science Foundation the Ministry of Education of China for an Innovative Research Team in University (IRT 17R88), and the Fundamental Research Funds for the Central Universities (Grant No. 31920190021)”.

We note that you have provided funding information that is not currently declared in your Funding Statement. However, funding information should not appear in the Funding section or other areas of your manuscript. We will only publish funding information present in the Funding Statement section of the online submission form.

“The funders had no role in study design, data collection and analysis, decision to publish, or preparation of the manuscript”.

Response: We are thankful for this suggestion. We apologize for our mistake, and we have deleted the funding information in the manuscript.

Response: We are thankful for your suggestion. No changes were required.

5. PLOS requires an ORCID iD for the corresponding author in Editorial Manager on papers submitted after December 6th, 2016. Please ensure that you have an ORCID iD and that it is validated in Editorial Manager. To do this, go to ‘Update my Information’ (in the upper left-hand corner of the main menu), and click on the Fetch/Validate link next to the ORCID field. This will take you to the ORCID site and allow you to create a new iD or authenticate a pre-existing iD in Editorial Manager. Please see the following video for instructions on linking an ORCID iD to your Editorial.

Response: Thank you for your positive response to our work and the kind advice. We have linked an ORCID iD to the Editorial Manager.

6. Please amend the manuscript submission data (via Edit Submission) to include author Xiaonong Guo.

Response: Thank you very much for these comments. We have amended the manuscript submission data.

7. Thank you for submitting the above manuscript to PLOS ONE. During our internal evaluation of the manuscript, we found significant text overlap between your submission and the following previously published works, some of which you are an author.

https://www.researchgate.net/publication/322729499_Complete_Chloroplast_Genome_Sequence_and_Phylogenetic_Analysis_of_Paeonia_ostii

https://www.mdpi.com/1422-0067/19/8/2443/htm

Please revise the manuscript to rephrase the duplicated text, cite your sources, and provide details as to how the current manuscript advances on previous work. Please note that further consideration is dependent on the submission of a manuscript that addresses these concerns about the overlap in text with published work.

Response: We apologize for our negligence and mistakes. We have checked throughout the text and corrected similar format issues. On the basis of the methods, we have reedited the contents and deleted the repetitive statements. Please see lines 111–134.

Responds to the reviewer’s comments:

Reviewer #1:

1. While the study appears to be sound, the language is unclear, making it difficult to follow. I advise the authors work with a writing coach or copyeditor to improve the flow and readability of the text.

Response: We have improved both the language and readability and sent our manuscript to LetPub for additional improvement of language and grammar.

2. The authors should carefully read and follow the PLOS ONE guidelines. There is no line number in the manuscript. Figures should be cited “Fig 1”, “Fig 2”, etc. References with more than six authors should list the first six author names, followed by “et al.”

Response: We apologize for our negligence and mistakes. We have checked throughout the text and corrected similar format issues. We added the line number in the revised manuscript. Thank you for pointing out this mistake in the figures. It has now been corrected. The reference format has also been corrected. Please see lines 387–388, 395–396, 519–522.

3. The authors should give more details for chloroplast genome assembly. How they defined the chloroplast-like reads? Please check the version of NOVOPlasty.

Response: According to your suggestion, we have described this method with more detail in the revised text. Please see lines 103–107. We have checked and corrected the version of NOVOPlasty. Please see line 99.

4. In Results, there are no results description in “Comparative Analysis of Genomic Structure” and “Phylogenetic Analysis”.

Response: Thank you for pointing this out. The mistake has now been corrected and this information has been added. Please see lines 185–196 and 206–219.

5. In Discussion, authors should avoid the first paragraph, they are part of the Introduction. And, authors need to emphasize the novel insights obtained from their study.

Response: Thank you for your advice. We modified the text according to your suggestion.

6. The authors said “used 13 chloroplast genome sequences for phylogenetic analysis” in Page 5, but there are 21 complete chloroplast genome sequences in Page 10 and Figure 3. Please check and correct it.

Response: We apologize for our negligence and mistakes. You are absolutely right about the number of chloroplast genome sequences for phylogenetic analysis, and we thank you for pointing out this mistake, which we have corrected. Please see line 136.

7. Which software was used to draw the phylogenetic tree? The authors said “used MEGA7.0 to analyze and draw a phylogenetic tree with ML (maximum likelihood)” in method. However, in Discussion was “the Bayesian Inference (BI) method based on RAxML was used to construct a phylogenetic tree”. Please carefully check.

Response: Thank you for pointing this out. We only used MEGA7.0 to analyze and draw a phylogenetic tree with ML (maximum likelihood). We have fixed the mistake at lines 274–278.

8. The reviewer suggested the authors to remove the Table 2.

Response: Thank you for your advice. We have removed it from our revised manuscript. Please see line 161.

9. In page 12, the authors mentioned the reference 74 in “Previous studies have reported that ycf3 is necessary for the stable accumulation of photosystem I complexes [42, 74]”, but missed it in References.

Response: Thank you for pointing out this mistake about the reference. According to the reviewer’s suggestion, we have added the mentioned literature in the reference section. Please see lines 544–546.

Reviewer #2: This manuscript “The chloroplast genome sequence of Apocynum venetum L.” reported the chloroplast genome sequence of Apocynum venetum, and provided some basic information about this genome, such as GC contents, LSC, SSC and coding gene information, which will be helpful for the studies in plant evolution. However, the written is very poor, and it is very descriptive that missing some very important information. For example, the author claim that the A. venetum chloroplast genome encodes 131 genes, including 86 protein-coding genes, 8 ribosomal RNA genes, and 37 transfer RNA genes. Is there any novel genes? How about this gene expression pattern, especially when plants meet the stress? The author could also compare the protein amount of chloroplast genes in some relative plant species and try to explain why Apocynum venetum could grow in such dry condition.

Response: Thank you very much for this insightful comment. The article has been polished by a professional editing company. At present, we could not detect novel genes in the chloroplast genome sequence of Apocynum venetum. In the future, we will excavate the stress-resistant genes and study their stress-resistance mechanisms in combination with transcriptome and physiological experiments.

---

## [Decision Letter · Decision Letter 1]

17 Nov 2021

PONE-D-21-20974R1The chloroplast genome sequence and phylogenetic analysis of Apocynum venetum L.PLOS ONE

Dear Dr. Guo,

Thank you for submitting your manuscript to PLOS ONE. After careful consideration, we feel that it has merit but does not fully meet PLOS ONE’s publication criteria as it currently stands. Therefore, we invite you to submit a revised version of the manuscript that addresses the points raised during the review process.

We look forward to receiving your revised manuscript.

Kind regards,

Maoteng Li

Academic Editor

PLOS ONE

Journal Requirements:

Reviewers' comments:

Reviewer's Responses to Questions

**Comments to the Author**

1. If the authors have adequately addressed your comments raised in a previous round of review and you feel that this manuscript is now acceptable for publication, you may indicate that here to bypass the “Comments to the Author” section, enter your conflict of interest statement in the “Confidential to Editor” section, and submit your "Accept" recommendation.

Reviewer #1: (No Response)

Reviewer #2: All comments have been addressed

2. Is the manuscript technically sound, and do the data support the conclusions?

Reviewer #1: Partly

Reviewer #2: Yes

3. Has the statistical analysis been performed appropriately and rigorously? 

Reviewer #1: Yes

Reviewer #2: Yes

4. Have the authors made all data underlying the findings in their manuscript fully available?

Reviewer #1: No

Reviewer #2: Yes

5. Is the manuscript presented in an intelligible fashion and written in standard English?

Reviewer #1: No

Reviewer #2: Yes

6. Review Comments to the Author

Reviewer #1: The new version of the manuscript has revised most of the comments and concerns that I rose in my previous review. However, there are many English language errors throughout the article. The author doesn't write seriously，there are many errors need to be corrected. Many references listed in the manuscript are not quoted accurately. For example, the references 50-53 and 54-56 in lines 129 and 131 respectively, these citations does not appear to have any context to the text. Moreover, the references are not numbered in the order in which they appear in the text. The cp genome sequence of A. venetum (MT568765) was not found in GenBank.

Reviewer #2: The authors have addressed all questions that I concerned. I recommend to accept this manuscript in this version.

7. PLOS authors have the option to publish the peer review history of their article (what does this mean?). If published, this will include your full peer review and any attached files.

Reviewer #1: No

Reviewer #2: No

---

## [Author Response · Author response to Decision Letter 1]

3 Dec 2021

Dear Editors and Reviewers:

Thank you for your letter and comments on our manuscript. These comments helped us improve our manuscript, and provided important guidance for future research.

We have addressed the editor’s and reviewers’ comments to the best of our abilities, and revised text to meet PLOS ONE style requirements. We hope this meets requirements for a publication.

The revised portions are marked in red in the manuscript. The main comments and our specific responses are detailed below:

Responses to the editor’s comments:

we feel that it has merit but does not fully meet PLOS ONE’s publication criteria as it currently stands. Therefore, we invite you to submit a revised version of the manuscript that addresses the points raised during the review process.

Response: Thank you very much for your comments, we carefully checked the format of references and improved references 2-4, 24 and 27, which are in lines 272-277 and 348-358 respectively; It was further verified that references 13 and 22 were adjusted interchangeably according to the citation, in lines 305-310 and 338-343, respectively, and reference 42 replaced a more appropriate reference. The details are in lines 400-405.

Response: The above three documents are named as required

Reviewer #1:

The new version of the manuscript has revised most of the comments and concerns that I rose in my previous review. However, there are many English language errors throughout the article. The author doesn't write seriously，there are many errors need to be corrected.

Response: Thanks for the constructive suggestions to improve our manuscript. We commissioned letpub, a professional editing agency, to edit the full text of the article, and attach the letpub editing certificate.

Many references listed in the manuscript are not quoted accurately. For example, the references 50-53 and 54-56 in lines 129 and 131 respectively, these citations does not appear to have any context to the text. Moreover, the references are not numbered in the order in which they appear in the text. 

Response: We are very sorry that some references were cited inappropriately due to our negligence. For 50-56 references, we selected the appropriate ones and replaced them one by one. See lines 427-462for details. secondly, we carefully checked the format of references and improved references 2-4, 24 and 27, which are in lines 272-277 and 348-358 respectively; It was further verified that references 13 and 22 were adjusted interchangeably according to the citation, in lines 305-310 and 338-343, respectively, and reference 42 replaced a more appropriate reference. The details are in lines 400-405.

The cp genome sequence of A. venetum (MT568765) was not found in GenBank.

Response: Since our article has not been published yet and the data has not been released yet, it cannot be searched on GenBank. This article is currently under minor revision. The data we communicate with ncbi will be released on December 6th.

Reviewer #2: 

The authors have addressed all questions that I concerned. I recommend to accept this manuscript in this version.

Response: Thank you very much for your comments.

---

## [Editor Report · Decision Letter 2]

9 Dec 2021

The chloroplast genome sequence and phylogenetic analysis of Apocynum venetum L.

PONE-D-21-20974R2

Dear Dr. Guo,

We’re pleased to inform you that your manuscript has been judged scientifically suitable for publication and will be formally accepted for publication once it meets all outstanding technical requirements.

Kind regards,

Maoteng Li

Academic Editor

PLOS ONE
---

## [Editor Report · Acceptance letter]

1 Mar 2022

PONE-D-21-20974R2 

The chloroplast genome sequence and phylogenetic analysis of *Apocynum venetum* L. 

Dear Dr. Guo:

I'm pleased to inform you that your manuscript has been deemed suitable for publication in PLOS ONE. Congratulations! Your manuscript is now with our production department. 

Kind regards, 

on behalf of

Dr. Maoteng Li 

Academic Editor

PLOS ONE